# Phylotranscriptomics unveil a Paleoproterozoic-Mesoproterozoic origin and deep relationships of the Viridiplantae

Zhiping Yang[1], Xiaoya Ma[1], Qiuping Wang[1], Xiaolin Tian[1], Jingyan Sun[1], Zhenhua Zhang[1], Shuhai Xiao ◉[2], Olivier De Clerck[3], Frederik Leliaert ◉[4] & Bojian Zhong ◉[1] ✉

The Viridiplantae comprise two main clades, the Chlorophyta (including a diverse array of marine and freshwater green algae) and the Streptophyta (consisting of the freshwater charophytes and the land plants). Lineages sister to core Chlorophyta, informally refer to as prasinophytes, form a grade of mainly planktonic green algae. Recently, one of these lineages, Prasinodermophyta, which is previously grouped with prasinophytes, has been identified as the sister lineage to both Chlorophyta and Streptophyta. Resolving the deep relationships among green plants is crucial for understanding the historical impact of green algal diversity on marine ecology and geochemistry, but has been proven difficult given the ancient timing of the diversification events. Through extensive taxon and gene sampling, we conduct large-scale phylogenomic analyses to resolve deep relationships and reveal the Prasinodermophyta as the lineage sister to Chlorophyta, raising questions about the necessity of classifying the Prasinodermophyta as a distinct phylum. We unveil that incomplete lineage sorting is the main cause of discordance regarding the placement of Prasinodermophyta. Molecular dating analyses suggest that crown-group green plants and crown-group Prasinodermophyta date back to the Paleoproterozoic-Mesoproterozoic. Our study establishes a plausible link between oxygen levels in the Paleoproterozoic-Mesoproterozoic and the origin of Viridiplantae.

The Viridiplantae, commonly known as green plants, including green algae and land plants, are a major group of photosynthetic eukaryotes that have played a prominent and uninterrupted role in the ecosystems globally at least since the Paleozoic[1,2]. Traditionally two main lineages have been recognized in the Viridiplantae, the Chlorophyta and the Streptophyta. Chlorophyta, initially diversified as unicellular planktonic algae mostly in marine environments[3], and gave rise to the modern prasinophytes and the morphologically and ecologically diverse core Chlorophyta[4,5]. Previous phylogenetic studies reported prasinophytes as a paraphyletic assemblage composed of several clades, which only recently were classified as separate classes, Mamiellophyceae, Pyramimonadophyceae, Nephroselmidophyceae, Palmophyllophyceae, Chloropicophyceae, and Picocystophyceae, as well as several environmental sequences lacking formal description[6–13] (Fig. 1). The traditional dichotomy of freshwater streptophytes and an early marine diversification of prasinophytes was recently shattered by phylogenomic analyses, which resolved Palmophyllophyceae and Prasinodermophyceae as a distinct phylum (Prasinodermophyta),

[1]College of Life Sciences, Nanjing Normal University, Nanjing, China. [2]Department of Geosciences and Global Change Center, Virginia Tech, Blacksburg, VA, USA. [3]Phycology Research Group and Center for Molecular Phylogenetics and Evolution, Ghent University, Ghent, Belgium. [4]Meise Botanic Garden, Meise, Belgium. ✉e-mail: bjzhong@gmail.com

**Fig. 1 | Current knowledge of phylogenetic relationships among the main lineages of Viridiplantae based on previous studies**[2,5,11,12,14,18,29]. Uncertain phylogenetic relationships are indicated by dashed lines. Drawings illustrate representative genera of each clade (not drawn to scale): (1) *Physcomitrella*, (2) *Mesostigma*, (3) *Tetraselmis*, (4) *Chlorella*, (5) *Chloropicon*, (6) *Picocystis*, (7) *Pycnococcus*, (8) *Pseudoscourfieldia*, (9) *Nephroselmis*, (10) *Pyramimonas*, (11) *Ostreococcus*, (12) *Monomastix*, (13) *Dolichomastix*, (14) *Prasinoderma*, (15) *Prasinococcus*, (16) *Verdigellas*.

diverging before the classic Chlorophyta and Streptophyta split[14]. Prasinodermophyta includes planktonic non-flagellate coccoids and benthic macroscopic species. Prasinophytes are also composed of planktonic non-flagellate coccoids and unicellular flagellates[15–20]. Prasinodermophytes and most prasinophytes live in marine environments and are globally distributed[21]. They are important contributors to primary productivity in marine systems and play a vital role in global elemental cycles[22–25]. For example, Mamiellophyceae and Chloropicophyceae (both in Chlorophyta) are major components of picoplanktonic communities in coastal and oligotrophic oceanic waters, significantly contributing to global carbon cycling[23,25–27]. In addition to their ecological significance, they represent excellent model systems for plant biology owing to their small-sized genomes with low gene redundancy[21,28]. Despite their non-negligible ecological significance and great value as model systems, the phylogenetic relationships and the origins of these green algae remain controversial.

The phylogenetic relationships within Prasinodermophytes and prasinophytes are difficult to resolve, as has been exemplified by the fact that different data types and analytical approaches have produced considerable nuclear-nuclear and plastid-nuclear discordance[29]. Plastid phylogenies consistently recovered the Prasinodermophyta as sister to Chlorophyta[11,29,30], while nuclear analyses yielded inconsistent positions concerning the Prasinodermophyta[14,29]. Nuclear rDNA analyses and multigene concatenated analyses generally supported the Prasinodermophyta as sister to all other green plants[11,14,29]. Conversely, in the coalescent analyses from 1KP initiative[29], Prasinodermophyta was allied with Streptophyta, whereas Li et al.[14] recovered Prasinodermophyta as sister to all other green plants. This incongruence may result from low taxon sampling, long-branch attraction (LBA), weak phylogenetic signal and/or various biological processes such as incomplete lineage sorting (ILS) and gene flow (both hybridization and introgression)[31–35].

Estimating the timeframe of Viridiplantae early divergences is important because the origin and early diversification of green plants set the stage for the transition from a cyanobacterial to a eukaryotic algal-dominated world[36]. Establishing the evolutionary timescale of Viridiplantae has been a difficult task, due to the poor fossil record of green algae. The sparse Proterozoic fossil record of green algae is further compromised by their simple morphologies and contentious interpretations. For example, acritarchs such as *Pterospermella* and *Tamanites* have been allied with *Pterosperma* and *Pachysphaera*, respectively, of the Pyramimonadophyceae[37–39]; however, these interpretations are uncertain as they rely on comparisons of simple morphological characters such as the presence of an annular membranous "wing" around the cell or pores through the cell wall[40–42]. Given the contentious taxonomic affinities of these Precambrian fossils, Nie et al.[30] found that the use of Precambrian fossils could generate older divergence-time estimates than generally accepted for crown-group green plants[30]. A well-preserved multicellular Precambrian fossil of green algae, *Proterocladus antiquus*, was found in the ca. 1.0 Ga Nanfen Formation from North China[43]. *P. antiquus* shares several morphological features with species of extant Cladophorales (Ulvophyceae), and has been interpreted as either a member of or as an extinct relative to the cladophoraleans, the ulvophyceans, the chlorophytes or the Viridiplantae[33,43]. Evaluation of different phylogenetic interpretations of this fossil are useful to quantify its effect on the time estimates of the origin and early diversification of the Viridiplantae[33,43]. Recently, a unicellular green alga fossil from the latest Ediacaran of South China, *Protocodium sinense*, has received attention for its taxonomic interpretation. This fossil can be conservatively used as a minimal calibration point for the origin of Bryopsidales[44], providing a new opportunity for reliable time estimates of the early diversification of the Viridiplantae. In general, fossils only provide a set of minimum age constraints which are insufficient for the dating analysis[45]. Therefore, a maximum age constraint on the root is particularly important in divergence-time estimation. The root maximum constraint can effectively restrict the age range of the entire tree. Different maximum constraints on the root may have large impacts on the same descendant clade age[45,46], which needs to be taken into account in dating analysis.

Biomarkers, or fossilized organic compounds derived from biomolecules, can also be used to constrain the age of green plant clades. Caution must be exercised, however, because biomarkers may not be clade-specific, can be contaminated by later sources or be affected by

preservational biases, and may reflect ecological dominance, rather than phylogenetic divergence[47]. Thus, biomarkers should be scrutinized for clade specificity, contamination, and preservational biases. Authentic biomarkers, like body fossils, should be regarded as minimum age constraints on clade divergence. With these caveats in mind, we note that early reports of eukaryotic biomarkers from Archean sediments[48] were later determined to be contaminants[49]. Instead, the abundant occurrence of eukaryotic biomarkers began in the Tonian Period around 820 Ma[50], although emerging data indicate that fossil sterols, which are mostly if not exclusively derived from eukaryotes, are abundant in Mesoproterozoic sediments 1.6–1.0 Ga[51]. Relevant to this study, the occurrence of abundant green algal biomarkers in Cryogenic rocks around 660 Ma[1,36,50] sets a minimum age for the divergence of Viridiplantae.

In this study we apply concatenation- and coalescent-based approaches to reconstruct deep relationships in the green plant phylogeny, and explore the causes for deep discordance concerning the phylogenetic position of Prasinodermophyta. Additionally, we estimate the timescale for Viridiplantae evolution by implementing four fossil calibration strategies based on different interpretations of *P. antiquus* and different maximum-age root constraints.

## Results and discussion
### Phylogenomic analyses of green plants
Our nuclear dataset consisted of 557 single-copy orthologous genes (SCOGs) derived from 68 genomes and transcriptomes. Phylogenetic relationships inferred from 557 SCOGs using concatenation-based (site-heterogeneous LG + C20 + F + G model) and coalescent-based approaches were highly congruent at the class and order level of green plants. However, certain discrepancies were observed for the positions of Picocystophyceae and Pseudoscourfieldiales relative to the other prasinophytes, as well as the position of Bryopsidales with respect to Ulvophyceae and Chlorophyceae, and Pedinophyceae and Chlorodendrophyceae relative to the other core Chlorophyta (Fig. 2a, b, Supplementary Fig. 1). Notably, the Prasinodermophyta consistently emerged as the sister clade to Chlorophyta with strong support (local posterior probability, LPP = 0.97) in the coalescent species tree (Fig. 2b). Likewise, the concatenation analyses using site-heterogeneous and site-homogeneous models all resolved Prasinodermophyta as the sister group to Chlorophyta, with Streptophyta as their common sister group (SH-aLRT = 100, BS = 100) (Fig. 2a, Supplementary Figs. 1 and 2). Our results deviate from the nuclear phylogeny reported by Li et al.[14], which showed that the Prasinodermophyta diverged prior to the split between the Chlorophyta and Streptophyta, suggesting the need to establish a separate phylum. In contrast, our phylogenomic analyses indicate that a three-phylum classification for the Viridiplantae may not be necessary, and that instead the traditional two-phylum model could be retained. The determination of whether to recognize one or two classes (Palmophyllophyceae and Prasinodermophyceae) is primarily a matter of taxonomic preference. However, considering the unresolved position of *Prasinococcus* within the clade (Supplementary Figs. 3 and 4) and the comparable genetic divergence observed within the Prasinodermophyta and the Mamiellophyceae, it may be preferable to adopt a single class, Palmophyllophyceae (including *Prasinoderma, Prasinococcus, Palmophyllum*, and *Verdigellas*), as initially defined by Leliaert et al.[11].

Apart from the positions of Picocystophyceae and Pseudoscourfieldiales, the branching order of the prasinophyte grade was identical in the concatenation and coalescent trees (Fig. 2a). In the concatenation tree, navigating the topology from the root towards the tips, we can successively identify a Pyramimonadophyceae + Mamiellophyceae clade, the Nephroselmidophyceae, a Pseudoscourfieldiales + Picocystophyceae clade, and Chloropicophyceae sister to core Chlorophyta, consistent with current understanding of prasinophyte phylogeny[2].

The Pseudoscourfieldiales + Picocystophyceae clade here recovered differs from the nuclear and plastid rDNA concatenated phylogeny of Li et al.[14]. This discrepancy may be attributed to the expanded sampling of Picocystophyceae and prasinophytes in our study. Yet, our coalescent analysis recovered Picocystophyceae as sister to a Chloropicophyceae + core Chlorophyta clade (Fig. 2b), consistent with coalescent tree inference of 1KP initiative[29]. In addition, our coalescent analysis weakly supported Pseudoscourfieldiales as sister to a Picocystophyceae + Chloropicophyceae + core Chlorophyta clade (LPP = 0.33) (Fig. 2b), while the coalescent analysis of 1KP initiative[29] weakly supported Pseudoscourfieldiales as sister to a Pyramimonadophyceae + Mamiellophyceae clade.

In addition, we compiled a plastid dataset comprising 74 plastid genes from 63 genomes. Phylogenetic relationships within prasinophytes were similar across different models (the site-homogeneous and site-heterogeneous models) except for the position of Pseudoscourfieldiales (Supplementary Figs. 3 and 4). Plastid topologies consistently recovered Prasinodermophyta as sister to Chlorophyta with strong support (SH-aLRT ≥ 95, BS ≥ 95) (Fig. 2a, Supplementary Figs. 3 and 4), consistent with our nuclear topologies and previously reported plastid topologies[11,29].

### Conflicting signal
The statistics of gene tree-coalescent tree and quartet tree discordance for each internal branch in the coalescent tree based on the 557 SCOGs dataset indicated rampant gene tree conflict at the deep internal branches (backbone) of our phylogeny (Supplementary Figs. 5 and 6), including the ancestral branch consisting of Prasinodermophyta and Chlorophyta (indicated with red asterisks in Fig. 2a, b). This focal branch associated with the position of Prasinodermophyta exhibited a high proportion of conflicting bipartitions (94.25%), while the concordant proportion was remarkably low (2.51%) (Fig. 2b). This conflicting phylogenetic signal regarding the position of the Prasinodermophyta was observed despite consistent recovery of Prasinodermophyta as sister to Chlorophyta in both concatenation and coalescent trees. To gain further insight into the distribution of phylogenetic signal in 557 SCOGs dataset, we quantified phylogenetic signal for each nuclear gene with respect to three alternative hypotheses for the position of the Prasinodermophyta. The fraction of loci strongly supporting Prasinodermophyta as sister to Chlorophyta (T1) was larger (19.89%) than the fraction of loci strongly supporting the other two alternative topologies (T2, 19.52% and T3, 9.76%) (Supplementary Fig. 7). Additionally, we conducted statistical analyses on the subsets of genes that strongly or weakly supported T1, T2, or T3, focusing on the distribution of parsimony informative sites, evolutionary rates, and saturation. Median values for parsimony informative sites showed that genes with strong signal (median value = 229–235) possessed a higher number of informative sites compared to those with weak signal (median value = 176.5–219) (Supplementary Fig. 7). In addition, both genes with strong signal (median value = 1.46–1.61) and genes with weak signal (median value = 1.44–0.47) exhibited rapid evolutionary rates. Similarly, there were non-negligible levels of substitution saturation in genes with both strong signal (median value = 0.14–0.15) and weak signal (median value = 0.15–0.16) (Supplementary Fig. 7). Thus, a large proportion of genes with weak signal (50.83%) in our dataset, as previously mentioned, can be attributed to a smaller number of informative sites, rather than higher levels of saturation or faster evolutionary rates.

### Potential causes of phylogenetic discordance regarding the position of Prasinodermophyta
Quartet frequencies (q1, q2 and q3) surrounding the focal branch (the ancestral branch consisting of Prasinodermophyta and Chlorophyta) showed the high extent of symmetry in two minor quartet frequencies

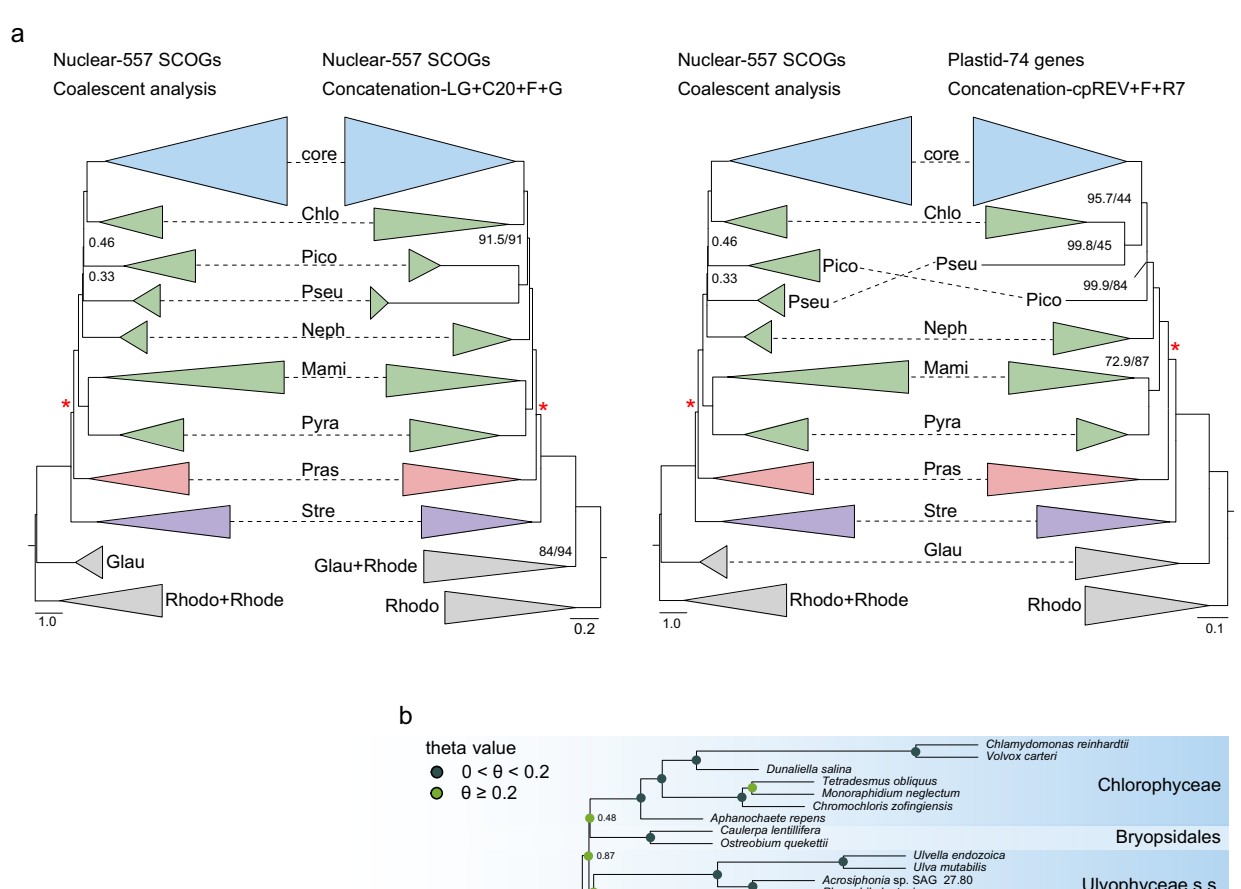

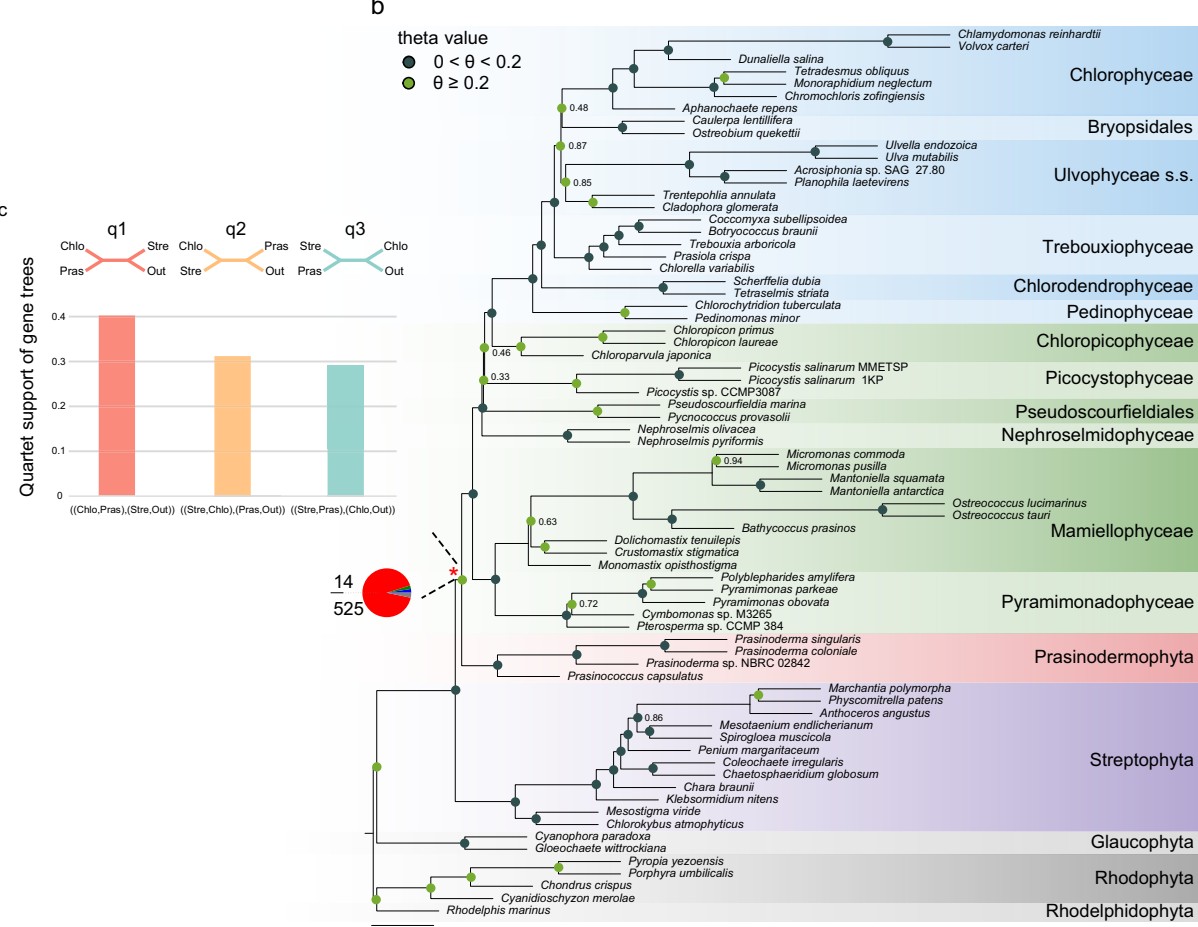

(q2 and q3). This finding suggests that ILS may have occurred in the early evolutionary history in green plants (Fig. 2c). Given that ILS can be the underlying cause of gene tree conflict regarding the position of Prasinodermophyta, we calculated the theta value of each internal branch based on a 72-taxon dataset of 557 SCOGs. The theta value of this focal branch (0.29) exceeded the average theta value of all

internal branches (0.21), indicating that ILS cannot be disregarded (Fig. 2b).

To further explore the topological variability of phylogenetic trees, we explored landscapes (topological variability) of phylogenetic trees and performed coalescent simulations based on a 7-taxon data-set (Fig. 3a) due to computational burden. In the landscape analysis of

**Fig. 2 | Tree reconstruction of the Viridiplantae. a** Topological comparison between coalescent and concatenation analyses based on nuclear and plastid datasets. Support values are shown only for nodes receiving less than 95% LPP (in coalescent-based analysis) and SH-aLRT/BS (in concatenation-based analyses). Chlo Chlorodendrophyceae, core core Chlorophyta, Pico Picocystophyceae, Neph Nephroselmidophyceae, Pseu Pseudoscourfieldiales, Mami Mamiellophyceae, Pyra Pyramimonadophyceae, Stre Streptophyta, Pras Prasinodermophyta, Glau Glaucophyta, Rhodo Rhodophyta, Rhode Rhodelphidophyta. **b** The phylogenetic tree constructed using ASTRAL based on 557 individual gene trees with 10% BS value cutoff. Support values are shown only for nodes receiving less than 95% support from LPP analysis. The pie chart at the ancestral branch of Prasinodermophyta and

Chlorophyta indicates the proportion of gene trees concordance and conflict against the coalescent species tree: blue, the proportion of gene trees supporting the species tree; green, the proportion of gene trees supporting the most common conflicting topology; red, the proportion of gene trees supporting all other conflicting topologies; dark gray, the proportion of the uninformative gene trees; light gray, the proportion of missing data. The color of circle point on each node represents two different intervals of theta values. The red asterisks in **a** and **b** indicate the ancestral branch consisting of Prasinodermophyta and Chlorophyta. **c** The quartet frequency (q1–q3) around the focal branch of ASTRAL tree based on the 72-taxon dataset. Each internal branch with four neighboring branches leads to three possible quartets.

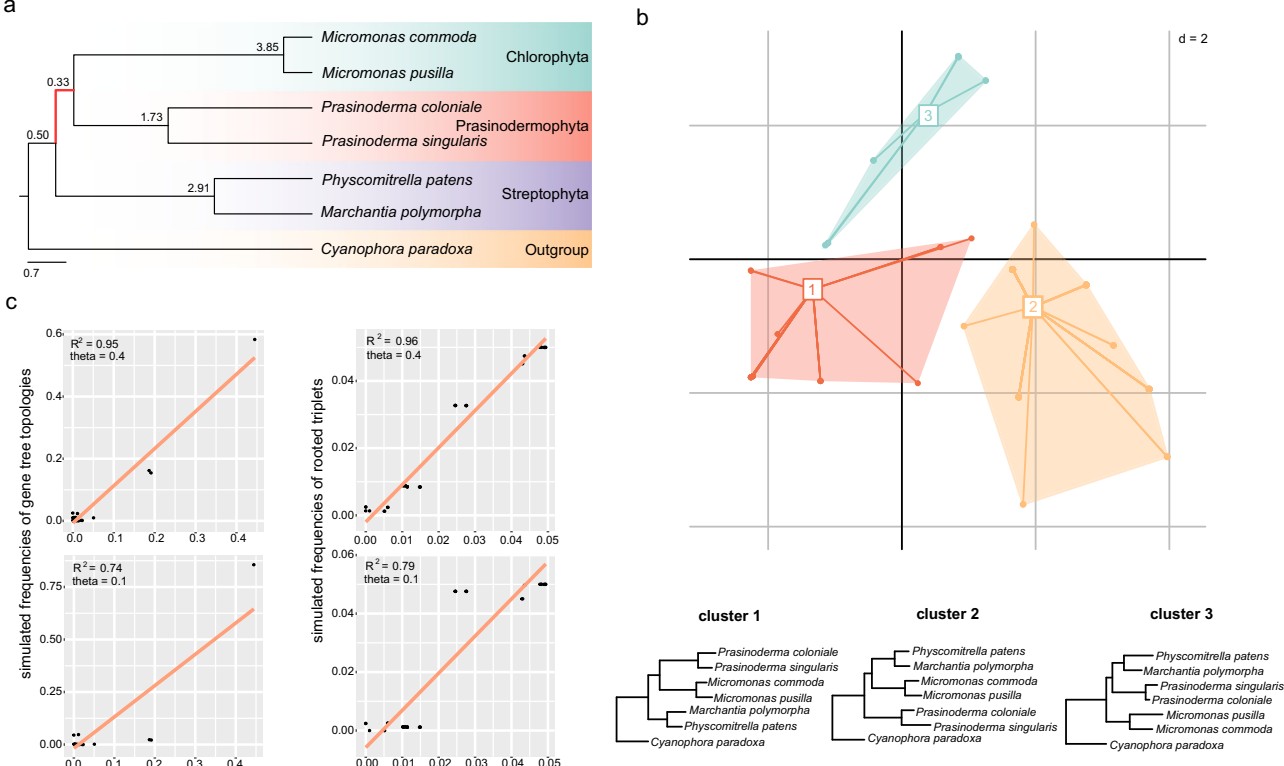

**Fig. 3 | The coalescent simulations and exploration of landscapes of phylogenetic trees. a** Phylogenetic relationships of a 7-taxon dataset based on the multispecies coalescent model. The short ancestral branch of Prasinodermophyta and Chlorophyta is highlighted in red. **b** Two-dimensional metric multidimensional scaling (MDS) plot of 434 gene trees based on the 7-taxon dataset. A consensus tree

of each cluster in MDS plot is displayed below. **c** Comparison of simulated frequencies of gene tree topologies and rooted triplets with the corresponding observed frequencies. The two figures above and below show the correlations obtained by setting theta value to 0.4 and 0.1 on the ancestral branch consisting of Prasinodermophyta and Chlorophyta, respectively.

phylogenetic trees, when the number of clusters was set to 3, the maximum clade credibility (MCC) tree (or the consensus tree) topology of clusters 1, 2, and 3 represented three alternative hypotheses regarding the position of Prasinodermophyta, respectively (Fig. 3b). This analysis showed that three alternative hypotheses for the position of the Prasinodermophyta were evident in terms of topological variability.

Furthermore, we performed coalescent simulations by generating 20,000 gene trees under high ILS conditions in Phybase and the multispecies coalescent model. The topological frequencies and triplet frequencies of simulated gene trees were in agreement with the empirical gene trees ($R^2 = 0.95/0.96$) (Fig. 3c). To assess the explanatory power of the coalescent model for the observed gene tree heterogeneity in the 7-taxon dataset, we calculated unweighted RF distances for empirical gene trees and simulated gene trees with high ILS conditions (theta = 0.4). The results indicated that the coalescent model accounted for 66% of gene tree variation observed for 434 gene

trees under high ILS conditions (Supplementary Fig. 8). We also simulated 20,000 gene trees with a low level of ILS (theta = 0.1) for the same dataset, but these simulated gene trees exhibited a lower correlation ($R^2 = 0.74/0.79$) with the empirical gene trees (Fig. 3c). These results indicate the ILS incorporated in the coalescent model could considerably account for gene tree discordance, especially the presence of two alternative topologies (T2 and T3). We therefore conclude that ILS is a major factor driving the gene tree conflict concerning the position of the Prasinodermophyta.

In addition to ILS, gene flow can also introduce gene tree discordance and lead to inaccurate species tree estimation[52,53]. We applied two approaches to investigate the level of gene flow causing discordance regarding the position of the Prasinodermophyta in the presence of ILS. Firstly, we measured the difference between the frequencies of two discordant topologies (three possible unrooted topologies of four taxa include one concordant and two discordant topologies) by calculating the Quartet Differential (QD) value

**Table 1 | Information of the calibration nodes**

| Node | Clade | Fossils | Node Calibration | Time prior (My) | Reference |
|------|-------|---------|------------------|-----------------|-----------|
| 1–1 | Root | NA | Root | 1891-1174 | Ref. 68,117 |
| 1–2 | Root | NA | Root | 3500-1174 | Ref. 69,117 |
| 2 | Rhodophyta | *Bangiomorpha* | Rhodophyta crown | Min 1174 | Ref. 117 |
| 3 | Rhodophyta | Doushantuo red algae | Floridiophyceae stem | Min 600 | Ref. 118 |
| 4–1 | Viridiplantae | *Proterocladus antiquus* | Viridiplantae crown | Min 1000 | Ref. 43 |
| 4–2 | Ulvophyceae | *Proterocladus antiquus* | Ulvophyceae s.s. stem[33] | Min 1000 | Ref. 43 |
| 5 | Marchantiophyta-Bryophyta | *Riccardiothallus devonicus* | Marchantiophyta-Bryophyta crown | Min 405 | Ref. 119 |
| | | Oldest cryptospores | Marchantiophyta-Bryophyta crown | Max 514 | Ref. 120 |
| 6 | Trebouxiophyceae | *Botryococcus* sp. | Botryococcus stem | Min 298.75 | Ref. 121 |
| 7 | Ulvophyceae | *Protocodium sinense* | Bryopsidales crown | Min 541 | Ref. 44 |
| 8 | Chlorophyceae | *Scenedesmus bifidus* | Scenedesmaceae stem | Min 125 | Ref. 122 |

Node numbers refer to the calibration points in Supplementary Figs. 12 and 13.

in Quartet Sampling[54] (e.g., QD < 1 can be indicative of gene flow). Secondly, we performed the chi-square test of the two minor quartet frequencies between empirical gene trees and simulated gene trees with ILS (e.g., *P*-value > 0.1 can be indicative of ILS). For each empirical gene tree, we collapsed branches with BS support below 10% to reduce gene tree estimation error. The ancestral branch (number 52 in Supplementary Fig. 9) of Prasinodermophyta and Chlorophyta (*P*-value = 0.71, QD value = 0.97) showed no significant difference between the observed and simulated dataset with respect to the symmetry in the two minor quartet frequencies (*P*-value > 0.1), and no significant difference between the frequencies of two discordant topologies in the observed dataset (QD value > 0.8) (Supplementary Fig. 9).

The coalescent simulations, quartet sampling, and chi-square tests indicate an important contribution of ILS to phylogenetic discordance surrounding this focal branch. The occurrence of ILS across the short ancestral branch of Prasinodermophyta and Chlorophyta suggests that the ancestor of green plants underwent an ancient radiation event, leading to rapid divergence into the major clades. The phylogenomic analyses support the sister relationship between Prasinodermophyta and Chlorophyta, indicating that the classical two-phylum model of the Viridiplantae can be retained. Our analyses highlight the importance of exploring multiple potential factors contributing to phylogenetic discordance in phylogenomic analyses, particularly when dealing with deep nodes exhibiting substantial variation in gene trees.

### Key evolutionary events in the history of the Viridiplantae

Our inferred phylogenetic relationships provided a solid framework for tracing key evolutionary events in green plant lineages, including freshwater-marine (or freshwater-terrestrial) transitions, the evolution of complex cyto-morphologies, and the loss of flagella (Supplementary Figs. 10 and 11, Supplementary Table 1). Ancestral state estimation suggested that the ancestor of green plants was most likely a unicellular freshwater alga with flagella, or at least with a flagellate stage in its life cycle, consistent with earlier conclusions drawn from chloroplast phylogenomic analyses of Archaeplastida[55].

Transitions from freshwater to marine or terrestrial habitats occurred independently multiple times within the Chlorophyta and Streptophyta (Supplementary Fig. 10). Multicellularity emerged from multiple independent origins, and complex cyto-morphologies evolved repeatedly in the core Chlorophyta (Supplementary Fig. 10), likely from unicellular ancestors, in agreement with previous analyses[56,57]. While the majority of green algae either possess flagella or exhibit flagellate stages in their life cycle, certain groups such as the Prasinodermophyta (Palmophyllophyceae and Prasinodermophyceae), some prasinophytes (e.g., Mamiellophyceae: *Ostreococcus* and *Bathycoccus*, Chloropicophyceae, Picocystophyceae, and *Pycnococcus*)

and some lineages of Trebouxiophyceae and Chlorophyceae, have never been reported to have flagellate stages[12,16,58–61]. Our results indicate that the loss of flagella occurred independently in these lineages, as well as certain lineages of the Streptophyta (Supplementary Fig. 11). However, these results obtained from the ancestral state estimation should be interpreted with caution.

Although flagella were most likely present in a life cycle stage of the green plant ancestor, it is plausible that this ancestor was a non-motile unicellular organism with transient flagellate stages[18]. This hypothesis finds support in the identification of numerous genes encoding proteins involved in flagellar structure within the genomes of *Prasinoderma* (Prasinodermophyta) and *Chloropicon* (Chloropicophyceae), implying the possible existence of cryptic flagellate stages in these species[14,62,63].

### The evolutionary timescale of green plants

It is worth mentioning that a number of organic-walled microfossils or acritarchs found in sedimentary rocks dating back approximately 1.8–2.0 Ga have been interpreted by some researchers as green algae[41,64], although diagnostic green algal features are lacking[65]. Several acritarch genera (e.g., *Tasmanites*, *Valeria*, and *Pterospermopsimorpha*) have been interpreted as green microalgae or even as prasinophytes[39,66]. If these interpretations are correct, then the occurrence of these acritarchs in 1.6–1.8 Ga rocks indicates that the divergence of green plants and prasinophytes may extend as the Paleoproterozoic[66,67]. It is important to note that the green algal interpretation of these acritarchs remains uncertain due to their relatively simple morphologies and the possibility of convergent evolution. Consequently, it is critical to establish a timescale of green algal evolution using molecular data, as we have attempted in this study.

To estimate a timeframe for the evolution of green plant lineages, we conducted molecular dating analyses employing four calibration strategies. Strategies 1 and 2 set the maximum constraint for the root age at 1.89 Ga[68] and placed the *P. antiquus* fossil at either node 4–1 or node 4–2 (Table 1, Fig. 4, Supplementary Fig. 12). In contrast, strategies 3 and 4 differed from the first two by setting the root maximum constraints to 3.50 Ga, representing the oldest known fossil evidence of life on Earth[69] (Table 1, Fig. 4, Supplementary Fig. 13). For each strategy, we compared the posterior distribution and the effective prior distribution of different calibrated nodes. Most of the fossil-calibrated nodes showed a significant shift of the posterior relative to the effective prior, indicating that the molecular data could provide information (Supplementary Fig. 14). By comparing divergence time estimates between strategies 1 and 2 (or between strategies 3 and 4), we found that the placement of the *P. antiquus* fossil significantly influenced the age estimates of most nodes. For strategies 1 and 2, the maximum value of the estimated 95% highest posterior density (HPD) interval of

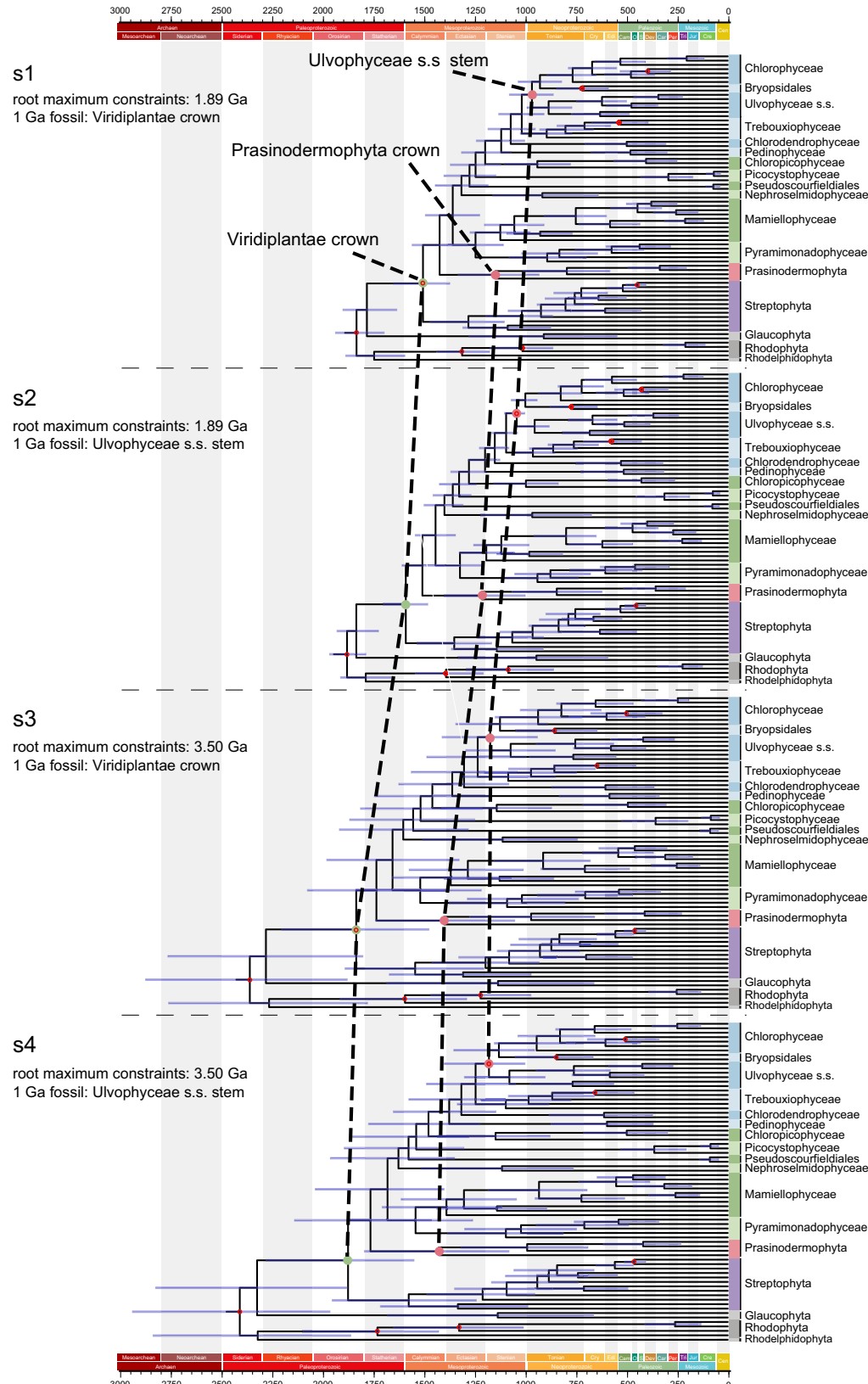

**Fig. 4 | Comparison of divergence time estimates from four fossil assignment strategies.** Node ages are plotted at the posterior means, with horizontal bars representing 95% credibility intervals ($n = 10,000$ samples for the MCMC process). The calibration nodes are represented by red hollow circles. The 1 Ga fossil refers to *Proterocladus antiquus*.

the root age all exceeded 1.89 Ga (Fig. 4, Supplementary Table 2), suggesting that older root maximum constraints may better account for the uncertainties. Therefore, our time estimation exploration suggests that green plants originated somewhere between the early Paleoproterozoic and early Mesoproterozoic (2.28–1.48 Ga) based on strategies 3 and 4. Similarly, strategies 3 and 4 provided divergence-time estimates for crown-group Prasinodermophyta ranging from the late Paleoproterozoic to late Mesoproterozoic (1.79–1.05 Ga). The origin of prasinophytes was also dated back to between the middle Paleoproterozoic and middle Mesoproterozoic (1.87–1.22 Ga), followed by a Mesoproterozoic origin of core Chlorophyta (1.65–1.08 Ga) (Supplementary Table 2). In addition to macrofossils like *P. antiquus* and *P. sinense* with specific taxonomic explanations, there are well-preserved fossils from the Dolores Creek Formation (0.95–0.90 Ga) that exhibit characteristics of green macroalgae[70–72]. Moreover, the macrofossil *Horodyskia* from the Tonian successions (0.95–0.72 Ga) in North China was tentatively interpreted as a multinucleate (coenocytic) green macroalga[73]. These fossils collectively indicate the early diversification of the Ulvophyceae, which can be traced back to at least the early Neoproterozoic, thus supporting our estimated origin of the Ulvophyceae s.s. (1.30–0.85 Ga).

## Geochemical evidence for the origin of green plants
Our molecular timeline presents older age estimates for crown-group green plants compared to previous large-scale analyses of eukaryotes[74], as well as studies focusing on land plants[30] or green algae[57]. The divergence of eukaryotes has traditionally been associated with the increase of atmospheric oxygen levels[75], although the causality of this relationship remains a topic of debate[76]. Geochemical data reveal two episodes of stepwise increase in atmospheric oxygen levels, known as the Great Oxidation Event at 2.3 Ga and the Neoproterozoic Oxygenation Event, c. 850–542 Ma[77]. During the intervening interval, atmospheric oxygen levels are believed to have been low, ranging between 0.1 to 1% of the present atmospheric level (PAL)[78,79], although some studies suggest higher atmospheric oxygen levels of 1–10% PAL during the Mesoproterozoic[80–84]. Oxygen levels between 1% and 10% PAL would certainly fulfill the respiration requirements of eukaryotic algae[85]. Even at the lower end of atmospheric oxygen levels (0.1–1% PAL), localized oxygen-rich oases may have existed in the surface ocean to support eukaryotic life[86]. Considering the oxygen availability in the Paleoproterozoic-Mesoproterozoic oceans, it is plausible that these environments were hospitable to green algae. Consequently, our estimated divergence times for crown-group green plants (2.28–1.48 Ga) and crown-group Prasinodermophyta (1.79–1.05 Ga) align well with geochemical evidence, which suggest that the oceans of the Paleo- and Mesoproterozoic provided a suitable habitat for early Viridiplantae, considering the surface redox conditions of the Earth.

## Methods
### Taxon sampling and orthology inference
Taxon sampling of the nuclear dataset was sourced from 35 publicly available genomes and 37 transcriptomes, covering all classes of Prasinodermophyta, Chlorophyta, Charophyta, and Bryophyta, as well as two glaucophytes, four red algae and *Rhodelphis marinus* (Rhodelphidophyta) as the outgroups (Supplementary Data 1). We inferred single-copy orthologous genes (SCOGs) using OrthoFinder v2.4.1[87] from eight published genomes of prasinophytes (*Micromonas commoda* and *Chloropicon primus*), prasinodermophytes (*Prasinoderma coloniale*), core chlorophytes (*Chlamydomonas reinhardtii*, *Chlorella variabilis*, and *Ulva mutabilis*) and streptophytes (*Physcomitrella patens* and *Marchantia polymorpha*). Then, HMMER v3.3.1[88] (with parameter -T 50) was used to identify 677 putative SCOGs from 64 additional genomes and transcriptomes. Redundancy in transcriptomes was reduced using CD-HIT v4.8.1[89] (with parameters -c 0.99

and -l 99). Each SCOG was aligned with the L-INS-I algorithm in MAFFT v7.471[90] and poor alignments with pairwise identity below 20% were removed. Then, each remaining alignment was pruned using trimAl v1.4[91] with the heuristic automated method. In order to reduce the influence of long branch attraction on the nuclear dataset, TreeShrink v1.3.2[92] was used with a threshold of 0.05 to identify abnormal sequences that resulted in unrealistically long branch lengths within each gene. Each SCOG with abnormal sequences removed was re-aligned and re-trimmed using MAFFT and trimAl. Besides, to reduce the impact of missing data on phylogenomic analyses, sequences in the trimmed alignments with taxon occupancy below 50% and lengths less than 100 amino acids were filtered out, resulting in a dataset of 557 SCOGs selected from 72 genomes and transcriptomes for the downstream phylogenomic analyses.

Taxon sampling of the plastid dataset included 63 plastid genomes downloaded from NCBI-GenBank, covering all classes of Prasinodermophyta, Chlorophyta, Charophyta, and Bryophyta, as well as two glaucophytes and three red algae as the outgroups (Supplementary Data 1). The amino acid sequence of each plastid gene was aligned with the L-INS-I algorithm in MAFFT. Then each alignment was pruned using trimAl[91] with the heuristic automated method. The influence of long branch attraction on the plastid dataset was reduced by TreeShrink with a threshold of 0.05. A plastid dataset of 74 genes with taxon occupancy above 50% was ultimately generated to reconstruct the plastid phylogeny.

### Tree reconstruction
Multispecies coalescent model and concatenation approaches were used to infer phylogenetic trees based on the dataset of 557 SCOGs. To estimate a coalescent-based tree, we firstly inferred 557 individual gene trees using RAxML v8.2.11[93] with the best-fit substitution model, and 100 rapid bootstrap (BS) replicates for clade support. In order to reduce the influence of gene tree estimation error, branches in gene trees with bootstrap support below 10% were collapsed by Newick Utilities v1.6[94] prior to the coalescent-based tree inferred by ASTRAL v5.7.4[95]. Local posterior probabilities[96] (LPP) were used to assess clade support for the ASTRAL tree. For the concatenation approach, the 557 SCOG alignments were concatenated to reconstruct the maximum likelihood (ML) trees with three types of evolutionary models and two partitioning schemes fed to IQ-TREE v2.0.7[97]: (1) a single partition with the best-fit substitution model searched by IQ-TREE (parameter -MFP): the site-homogeneous LG + F + R10 model; (2) a single partition with the site-heterogeneous LG + C20 + F + G model; (3) a gene-wise partitioned analysis using the best-fit model estimated to each partition. Supports of each analysis were estimated with 1,000 ultrafast bootstrap and SH-aLRT branch test replicates[98].

For the plastid dataset, 74 plastid gene alignments were concatenated into a supermatrix and then applied to reconstruct concatenation trees with three types of evolutionary models by IQ-TREE: (1) a single partition with the best-fit substitution model searched by IQ-TREE: the site-homogeneous cpREV + F + R7 model; (2) a single partition with the site-heterogeneous cpREV + F + G + C20 model; (3) a gene-wise partitioned analysis using the best-fit model estimated to each partition. Supports of each analysis were estimated with 1000 ultrafast bootstrap and SH-aLRT branch test replicates.

### Detection of gene tree conflict and quartet tree discordance
We used PhyParts[99] to map individual gene trees on the coalescent tree with a bootstrap support (BS) threshold of 10% to examine patterns of gene tree-coalescent tree concordance and conflict within the nuclear dataset of 557 SCOGs. The pie chart of each internal branch (node) was summarized and generated by PhyPartsPieCharts and ETE3[100] (https://github.com/mossmatters/MJPythonNotebooks). For the same purpose of an interpretable measure of gene-tree heterogeneity specific to each internal branch, the quartet frequency around each

internal branch of the coalescent tree reconstructed by 557 SCOGs was summarized and generated by ASTRAL and AstralPlane v0.1.1 (https://github.com/chutter/AstralPlane) with a bootstrap support threshold of 10%.

### Dissection of phylogenetic signal

Gene-wise log-likelihood scores among the three constraint topologies (T1, T2, and T3) for each gene in the dataset of 557 SCOGs were calculated to quantify the distribution of the phylogenetic signal over three alternative hypotheses. If a gene satisfied lnL(T1)-lnL(T2) ≥ 2 and lnL(T1)-lnL(T3) ≥ 2, this gene is supposed to strongly support T1. If a gene satisfied 0 <lnL(T1)-lnL(T2) < 2 and lnL(T1)-lnL(T3) ≥ 2, 0 <lnL(T1)-lnL(T2) < 2 and 0 <lnL(T1)-lnL(T3) < 2, or lnL(T1)-lnL(T2) ≥ 2 and 0 <lnL(T1)-lnL(T3) < 2, this gene is supposed to weakly support T1. By analogy, we then identified other genes that strongly or weakly support T2 or T3. The number of genes that strongly or weakly support one of three topologies was calculated. We also explored the correlation of the strength of the gene signal with parsimony informative sites, substitution saturation, and evolutionary rates. Calculations of substitution saturation and evolutionary rates are based on TreSpEx v1.1[101]. The level of saturation is measured by the slope of the patristic distances (PD) calculated from the tree against the uncorrected genetic distance (p) calculated from the alignment. The evolutionary rate of each gene is derived from the average PD between any pair of taxa in the corresponding gene tree.

### Detection of ILS and gene flow

In order to explore if gene tree discordance, concerning the position of the Prasinodermophyta was related to incomplete lineage sorting, we implemented the following analyses: (1) We calculated the population size parameter θ of each internal branch based on the nuclear dataset of 557 SCOGs. A ML tree with branch lengths in mutational units (μT) was inferred with RAxML-NG v.1.2.0[102] by constraining a tree search to the ASTRAL tree topology and using a partition-by-gene scheme with best fit model for each partition (with parameters -evaluate and -brlen scaled). The population size parameter θ (θ = μ T/τ[103]) reflecting the ILS level for each internal branch was calculated from dividing the RAxML-NG mutational branch length by the ASTRAL coalescent branch length[35,104]. (2) For giving our main focus on identifying potential ILS among major clades of green plants, we explored phylogenetic tree landscapes and performed the coalescent simulation on the 7-taxon dataset. We reduced our taxon sampling to one outgroup (one species of the Glaucophyta) and six ingroup taxa (two species of the Prasinodermophyta, two species of the Chlorophyta, two species of the Streptophyta). After aligning, trimming and outlier removal with TreeShrink, 434 genes with a minimum of 100 aligned amino acid pairs and 100% taxa occupancy were retained for reconstructing individual RAxML gene trees. For the investigation of phylogenetic tree landscapes, the metric multidimensional scaling (MDS) plot and median tree plot were generated based on these rooted RAxML gene trees using the function findGroves (with the parameter method = "treeVec", nf = 4, nclust = 3) and the function med.trees in TREESPACE v1.1.4.2[105]. For the analysis of coalescent simulation, the population size parameter θ of each internal branch was calculated by the RAxML-NG tree and ASTRAL tree, and the theta value for terminal branches was set to a constant value of 1. Then, combining the constraint tree and the estimated theta values, we simulated 20,000 gene trees with the sim.coaltree.sp function in Phybase v.1.5[106]. We also simulated 20,000 gene trees with a low ILS level (set the theta value on the ancestral branch composed of Prasinodermophyta and Chlorophyta to 0.1). The topological frequencies of simulated gene trees and empirical gene trees were calculated. In addition to topological frequencies, we summarized the triplet frequencies from gene trees. Rooted triplets were generated by pruning all possible combinations of three ingroup taxa plus an outgroup from each gene tree with ETE3[100], and triplet frequencies of simulated gene trees and empirical gene trees were calculated. Lastly, the correlation tests were performed for topological frequencies and triplet frequencies using the cor.test function in R. To estimate how much ILS contributes to gene tree heterogeneity observed, we calculated unweighted RF distances for empirical gene trees and simulated gene trees in the 7-taxon dataset.

Furthermore, we explored whether ILS or gene flow was a main cause of discordance concerning the position of Prasinodermophyta. Based on the theta value of each internal branch and the RAxML-NG tree (the constraint tree) inferred from the nuclear dataset of 557 SCOGs, 20,000 gene trees were simulated using the sim.coaltree.sp function in Phybase v.1.5. The theta value for terminal branches was set to a constant value of 1. The two minor quartet frequencies (quartet frequencies of two minor topologies) of 557 empirical gene trees and 20,000 simulated gene trees with ILS were calculated by the parameter "-t 32" in ASTRAL. Then, a two-sided chi-squared test was performed on those quartet frequencies. We also used Quartet Sampling v1.3.1 (https://github.com/fephyfofum/quartetsampling) to calculate Quartet Differential (QD) score which measures the difference between the two minor quartet frequencies of 557 empirical gene trees. Finally, it is determined whether a significant level of ILS has been observed in the focal ancestral branch related to the position of Prasinodermophyta based on the criteria of whether P-value greater than 0.1, QD value greater than 0.8, similar to Ma et al.[107] and Suvorov et al.[108].

### Ancestral state reconstruction

We coded habitat, flagellate, and morphological characters of 72 species for ancestral state reconstruction (Supplementary Table 1). Habitat characters include freshwater, terrestrial, and marine. Morphological characters include unicellular, colonial, multicellular, siphonous, and siphonocladous. Flagellate characters refer to whether flagellate stages can be observed in the life cycle. The ultrametric trees were used to guide the ancestral state reconstruction of three traits (habitat, flagellate stages and cell morphology) in phytools v1.5.1 R package[109]. We estimated the discrete character evolution with ARD model which allows every possible type of transition has a different rate. The posterior probabilities of each character for each ancestral node were calculated from summaries of 1000 simulations of stochastic character mapping under ARD model (the function make.simmap).

### Divergence time estimation

**Dataset assembly for molecular dating.** The most clock-like 100 genes were identified from the dataset of 557 SCOGs according to the priority order of the bipartition support, root-to-tip variance, and tree length by SortaDate[110]. Selecting the most clock-like genes could minimize the impact of model misspecification for the estimation of divergence times.

**Rate priors.** A relaxed molecular clock model that accounts for rate variation among lineages was set for divergence time estimation. We calculated the amino acid pairwise distance between *Caulerpa lentillifera* and *Ulva mutabilis* with LG + Γ4 + F model by the package CODEML in PAML package v4.9j[111]. According to *P. antiquus* fossil placement, the divergence time between the two species was about 1.0 Ga[43], which indicated that the mean rate for branches (μ) was assigned to a gamma hyperprior G (2, 50.28) with 2/50.28 = 0.04 substitutions per site per time unit (100 My). The rate drift parameter was assigned gamma hyper-prior σ2 ~ G (1, 10) with the mean 0.1.

**Time priors.** The time prior depends on the information of the fossil calibration point and the birth-death process. Birth-death process includes three parameters: birth rate (λ), death rate (μ), and sample fraction (ρ). The ddBD tree prior was used to calculate the birth rate and death rate because it can produce accurate node ages and confidence

intervals even with a few well-constrained calibrations[112]. The sampling fraction was the proportion of our ingroup sample size (65 taxa) to the number of extant species in the, Prasinodermophyta, Chlorophyta, Charophyta (https://www.algaebase.org) and Bryophyta[113] (~33,413).

**Fossil constraints.** We estimated the evolutionary timeline of the Viridiplantae using four analyses: two different root soft maximum constraints (1.89 Ga and 3.50 Ga) × two different positions of *P. antiquus* fossil calibration (node 4–1 and 4–2). Not only are there few records of green algal fossils, but some Precambrian green algal fossils with simple morphology have low taxonomic reliability. Thus, eight fossil calibration points were carefully selected for divergence time estimation. According to phylogenetic justifications of these fossils, uniform priors with a hard minimum bound (pL = 1e$^{-300}$) and a soft maximum bound (pU = 0.025) were used for node 1 and 5, while cauchy priors with a hard minimum bound (pL = 1e$^{-300}$) were used for other nodes (Table 1).

**Approximate likelihood calculation and convergence assessment.** The approximate likelihood calculations in MCMCTree were implemented under the LG + Γ4 + F model using CODEML[114]. We performed two runs, each consisting of $1 \times 10^7$ iterations after a burn-in of $1 \times 10^6$ iterations and sampling every 1000. Both independent runs using Tracer v. 1.7.2[115] showed evidence of convergence (effective sample sizes, ESS values > 200). Calibration densities of the effective prior and posterior distributions were plotted by MCMCTreeR in R[116].

### Reporting summary
Further information on research design is available in the Nature Portfolio Reporting Summary linked to this article.

## Data availability
Data associated with the analyses are available from open-access repositories: https://figshare.com/s/a042722630e5ca0b2ba5. The data include sequence alignments, inferred phylogenies, simulated data, and time-calibrated trees. Taxon sampling of the nuclear and plastid datasets are provided in Supplementary Data 1.

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

## Acknowledgements
This work is supported by the National Natural Science Foundation of China (32122010 and 31970229), Collaborative Innovation Center for Modern Crop Production co-sponsored by Province and Ministry (CIC-MCP), and the Priority Academic Program Development of Jiangsu Higher Education Institutions (PAPD). We thank Lingxiao Yang, Zheng Hou, Alison Cloutier, Ya Yang and Diego Morales-Briones for helpful discussion.

## Author contributions
B.Z. conceived and designed the research; Z.Y., X.M., Q.W., X.T., J.S., and Z.Z. performed the experiments and analyzed the data; Z.Y. and B.Z. drafted the papers with input from all authors; Z.Y., S.X., O.D.C., F.L., and B.Z. revised and edited the paper. All of the authors read and approved the final paper.

## Competing interests
The authors declare no competing interests.
