## [Peer Review File · Nature Communications]

Phylotranscriptomics unveil a Paleoproterozoic-Mesoproterozoic origin and deep relationships of the ViridiplantaeReviewers' Comments:

Reviewer #1:

Remarks to the Author:

In this manuscript Yang and co-authors re-analyze available sequencing data of nuclear genomes from 68 species, as well as plastid genes from 60 species to analyze deep phylogenetic relationships of the Viridiplantae. Indeed, the traditional dichotomy between land plants and green algae has been recently questioned by the phylogenetic analysis including the newly sequenced genome of *Prasinoderma coloniale* (Li et al 2020), resolving Palmophyllophyceae as a distinct phylum, the Prasinodermophyta.

Here, the authors investigate available data and the effect of gene sampling, long branch attraction, incomplete lineage sorting (ILS) and gene flow, to understand the incongruence reported between previous phylogenetic studies on the position of Prasinodermophyta within the Chlorophyta or as an earlier diverging branch.

They also perform molecular dating analyses with different calibration strategies to infer divergence times between green plants and Prasinodermophyta.

I find their effort to sort out the effect of Incomplete lineage Sorting on the conflicting phylogenetic signals observed between gene families particularly interesting.

I enjoyed reading this well written manuscript and I found the figures superb, so that I have very few comments.

My main concern is that the methodological approach to disentangle the effect of ILS versus ancient hybridization is not sufficiently detailed.

1. ILS alone is not expected to result in different coalescence times between two discordant topologies, and I find it surprising that there is no report on introgression events in the manuscript. You conclude that ILS is the main cause of discordance concerning the position of Prasinodermophyta, but why is there no report of any introgression event? The very ancient candidate introgression events could be really interesting.

2. Line 393-394 : "The ILS level was estimated by dividing the PAUP mutational branch length by the ASTRAM coalescent branch length." Could you please add a reference about this metric or provide additional explanation about this estimation ?

3. Line 217-220. I find that the claim "the ancestor of green plants was a unicellular marine alga" is not necessarily in agreement with "Archaeplastida evolved in low-salinity environments and green algae colonized marine environments early in their histories", could you please clarify ?

Gwenael Piganeau

Reviewer #2:

Remarks to the Author:

The manuscript by Yang et al. resolved the deep relationships of early-branching lineages of green plants and showed ILS was the main cause of conflicts in the position of Prasinodermophyta. The molecular dating analyses tested different calibration strategies and estimated the crown-group green plants and crown-group Prasinodermophyta originated in the Paleoproterozoic-Mesoproterozoic. In addition, the geochemical evidence is largely consistent with the molecular dating analyses. This

manuscript in general is well written and presented with nice figures. The findings provide valuable insights into the evolution of green plants, which has general interests to the readership of the journal. There are several issues listed below.

Major comments:

1. The authors claimed that a large proportion of genes have weak phylogenetic signals due to short gene length or few informative sites. It would be better to show the exact number of genes with weak phylogenetic signals.
2. The recent paper (Chai et al., 2022, BMC Biology) reported *Codium*-like fossil (*Protocodium sinense* gen. et sp. nov.) from the latest Ediacaran. It might be helpful to use *Protocodium* for the origin of Bryopsidales as conservative calibration, in order to evaluate the impact on the estimation of the divergence time of early-branching lineages of green plants.
3. Line 206-208 : It is impressive that the authors considered hybridization as another major component of gene tree heterogeneity, but the related results are lacking. Please add detailed results about hybridization analyses.
4. For the section of "Key evolutionary events in the history of the Viridiplantae", it is interesting to trace the evolutionary history of flagella of early-diverging green plants. I suggest to add a figure showing the presence or absence of flagellate stages in the life cycle of green algae in the main text, rather than in supplemental information.
5. The authors suggested that ILS was the main cause of inconsistent phylogenetic results. Is there any possibility that the authors could discuss more about ILS related to possible ancient radiation or adaptative evolution to global climates?

Minor comments:

1. Line 79: "Proterocladus antiquus" -> "P. antiquus"
2. Line 122: "with fast-evolving genes removed" -> "without fast-evolving genes".
3. Line 121-123: Please use the same term "long-branch artefact" or "long-branch attraction artifact".
4. Line 132: "split in the Viridiplantae between Chlorophyta and Streptophyta" -> "split between Chlorophyta and Streptophyta".
5. Line 136: "Pseudoscourfieldiales (clade V)" -> "Pseudoscourfieldiales".
6. Line 203: e.g. -> e.g.,
7. Line 335 : "For the concatenation approach, The 576 OG alignments..."-> "For the concatenation approach, the 576 OG alignments...".
8. Line 478, Line 482: the reference should be checked carefully, especially the journal and author names.

Reviewer #3:

Remarks to the Author:

This manuscript adds to the body of evidence supporting Prasinodermophyta as sister to all other Viridiplantae groups. Understanding relationships across the green plant tree of life (and specially towards the root) is much needed to clarify, not just their role in key planetary events (e.g., great oxidation events during the Paleo- and Neoproterozoic), but also their evolution through time and how these events may have been caused by them or, instead, may have shaped them.

Additionally, this manuscript sets an example on how to carefully disentangle systemic bias from biological signal, by presenting an in-depth exploration of biological mechanisms and systematic errors

(methodological artifacts) underlying phylogenetic conflict and incongruence/discordance.

This manuscript also explores alternative timelines of events in early green plant diversification. The authors' constructive exploration of the effect alternative fossil constraints can have on ancestral divergence time estimation also adds to the literature trying to reconcile molecular clocks with the fossil and geochemical records.

For the most part, the authors' conclusions and claims are supported. I am missing, however, further exploration on the effect alternative outgroups can have in the ingroup topology, which could be having a major say on the long-branch attraction (LBA) detected (please, see further comments in the file attached).

In my opinion, further analyses with an extended outgroup that comprises Glaucophyta, Picozoa, and Rhodelphidia are needed before proceeding to accepting this work for publication (data can be mined from OA repositories). A major revision would be required, mostly because of the time repeating all of this tremendous work would take (with the extended outgroup mentioned above), not because of the quality of the analyses presented (which is excellent).

I would like to point out description of phylogenies could be further improved. Two clades (be them species poor or species rich) sharing a common ancestor (i.e., a node) are sister to each other (period). None of them is basal to the other, one just happens to have a lot less tips than the other (which could, for instance, result from much higher extinction rates, rather than lower speciation ones). Please, be mindful of terms such as "basal", "early", "primitive", etc., which can bias your readers' perceptions of the evolution of these organisms.

The methodology the authors present is sound and up to date (albeit the rudimentary character state reconstruction approaches implemented are rather limited on their reach). It meets the standards of the field; however, further work is needed to address the effect the outgroup chosen could be having on the LBA detected. I am also missing an extra step to remove outliers from gene alignments/topologies (see attached file for further details) prior to inferring species trees in a multispecies coalescent framework (it is known the quality of the input gene trees can greatly determine the output species tree topology). In all, there is sufficient level of detail to reproduce the authors' approach (for those familiar with the methods used and who have certain degree of prior bioinformatic knowledge, that is).

REVIEWER COMMENTS

Reviewer #1 (Remarks to the Author):

In this manuscript Yang and co-authors re-analyze available sequencing data of nuclear genomes from 68 species, as well as plastid genes from 60 species to analyze deep phylogenetic relationships of the Viridiplantae. Indeed, the traditional dichotomy between land plants and green algae has been recently questioned by the phylogenetic analysis including the newly sequenced genome of *Prasinoderma coloniale* (Li et al 2020), resolving Palmophyllophyceae as a distinct phylum, the Prasinodermophyta.

Here, the authors investigate available data and the effect of gene sampling, long branch attraction, incomplete lineage sorting (ILS) and gene flow, to understand the incongruence reported between previous phylogenetic studies on the position of Prasinodermophyta within the Chlorophyta or as an earlier diverging branch.

They also perform molecular dating analyses with different calibration strategies to infer divergence times between green plants and Prasinodermophyta.

I find their effort to sort out the effect of Incomplete lineage Sorting on the conflicting phylogenetic signals observed between gene families particularly interesting.

I enjoyed reading this well written manuscript and I found the figures superb, so that I have very few comments.

Reply: Thank you very much for your positive comments.

My main concern is that the methodological approach to disentangle the effect of ILS versus ancient hybridization is not sufficiently detailed.

Reply: We further detailed the methodological approach as follows: "Based on the theta value of each internal branch and the RAxML-NG tree (the constraint tree) inferred from the nuclear dataset of 557 SCOGs (single-copy orthologous genes), 20,000 gene trees were simulated using the sim.coal.tree.sp function in Phybase v.1.5. The theta value

for terminal branches was set to a constant value of 1. The two minor quartet frequencies of 557 empirical gene trees and 20,000 simulated gene trees with ILS were calculated by the parameter "-t 32" in ASTRAL v5.7.4. Then, a two-sided chi-squared test was performed on those quartet frequencies. We also used Quartet Sampling v1.3.1 (<https://github.com/fephyfofum/quartetsampling>) to calculate Quartet Differential (QD) score which measures the difference between the two minor quartet frequencies of 557 empirical gene trees".

1. ILS alone is not expected to result in different coalescence times between two discordant topologies, and I find it surprising that there is no report on introgression events in the manuscript. You conclude that ILS is the main cause of discordance concerning the position of Prasinodermophyta, but why is there no report of any introgression event? The very ancient candidate introgression events could be really interesting.

Reply: "Introgression" is the one way gene flow between species mediated by hybridization. Hybridization and introgression are the eventual outcome of gene flow. We used P-value in chi square test and QD value in Quartet Sampling to identify the level of gene flow in the presence of ILS. The result of the ancestral branch of Prasinodermophyta and Chlorophyta (P-value = 0.71, QD value =0.97) suggests a significant level of ILS and a low level of gene flow. Thus, we conclude that ILS is the main cause of discordance concerning the position of Prasinodermophyta from the perspective of biological process. This has been clarified in the manuscript.

2. Line 393-394 : "The ILS level was estimated by dividing the PAUP mutational branch length by the ASTRAM coalescent branch length." Could you please add a reference about this metric or provide additional explanation about this estimation ?

Reply: The population size parameter θ ($\theta = \mu T/\tau$; Degnan and Rosenberg, 2009) reflecting the ILS level for each internal branch was calculated from dividing the RAxML-NG mutational branch length by the ASTRAL coalescent branch length (Cloutier et al., 2019; Morales-Briones et al., 2020). We added references about this metric in the revised version.

3. Line 217-220. I find that the claim “the ancestor of green plants was a unicellular marine alga” is not necessarily in agreement with “Archaeplastida evolved in low-salinity environments and green algae colonized marine environments early in their histories”, could you please clarify ?

Reply: In the revised version of the manuscript, our analyses included more outgroups and revealed a likely freshwater origin of green plants which is congruent with the analyses of Sánchez-Baracaldo et al. (2017). We adjusted our corresponding expressions in the main text as follows: “Ancestral state estimation suggested that the ancestor of green plants was most likely a unicellular freshwater alga with flagella, or at least with a flagellate stage in its life cycle, consistent with earlier conclusions drawn from chloroplast phylogenomic analyses of Archaeplastida (Sánchez-Baracaldo et al., 2017)”.

Gwenaél Piganeau

Reviewer #2 (Remarks to the Author):

The manuscript by Yang et al. resolved the deep relationships of early-branching lineages of green plants and showed ILS was the main cause of conflicts in the position of Prasinodermophyta. The molecular dating analyses tested different calibration strategies and estimated the crown-group green plants and crown-group Prasinodermophyta originated in the Paleoproterozoic-Mesoproterozoic. In addition, the geochemical evidence is largely consistent with the molecular dating analyses. This manuscript in general is well written and presented with nice figures. The findings provide valuable insights into the evolution of green plants, which has general interests to the readership of the journal. There are several issues listed below.

Reply: Thank you very much for your positive comments.

Major comments:

1. The authors claimed that a large proportion of genes have weak phylogenetic signals due

to short gene length or few informative sites. It would be better to show the exact number of genes with weak phylogenetic signals.

Reply: *We added specific number of genes with weak phylogenetic signals in the revised version as follows: “Thus, a large proportion of genes with weak signal (50.83%) in our analysis results from fewer informative sites, rather than higher levels of substitution saturation or faster evolutionary rates”.*

2. The recent paper (Chai et al., 2022, BMC Biology) reported *Codium*-like fossil (*Protocodium sinense* gen. et sp. nov.) from the latest Ediacaran. It might be helpful to use *Protocodium* for the origin of Bryopsidales as conservative calibration, in order to evaluate the impact on the estimation of the divergence time of early-branching lineages of green plants.

Reply: *Considering the taxonomic interpretation of *P. sinense* is reliable, we added this fossil in dating analyses in the revised version. The updated dating analyses yield divergence-time estimates ranging from the Paleoproterozoic to Mesoproterozoic for crown-group Viridiplantae (2.28–1.48 Ga) and crown-group Prasinodermophyta (1.79–1.05 Ga). We also updated relevant descriptions and tables in the revised version.*

3. Line 206-208: It is impressive that the authors considered hybridization as another major component of gene tree heterogeneity, but the related results are lacking. Please add detailed results about hybridization analyses.

Reply: *We used the P-value and QD value to identify the cause of topological conflict (ILS or gene flow) around the internal branch. If P-value is greater than 0.1 and QD value is greater than 0.8, it is considered that the significant level of ILS is the main cause of phylogenetic discordance surrounding that internal branch, and the role of gene flow can be ignored (Ma et al., 2021; Suvorov et al. 2022). Thus, we argue that ILS is a major factor causing gene tree heterogeneity surrounding the ancestral branch (No.52) of Prasinodermophyta and Chlorophyta (P-value = 0.71, OQ value = 0.97).*

4. For the section of “Key evolutionary events in the history of the Viridiplantae” , it is interesting to trace the evolutionary history of flagella of early-diverging green plants. I suggest to add a figure showing the presence or absence of flagellate stages in the life cycle of green algae in the main text, rather than in supplemental information.

Reply: The ancestral state estimation is conditional on taxon sampling. We conducted intensive sampling in green plants, but sampled relatively sparse in red algae due to the lack of high-quality genomes of red algae. This may influence character estimation of some ancestral nodes. In addition, there are also limitations to the method of reconstructing ancestral states. For example, no software can select the most suitable model among all applicable models for a trait. Thus, we decide to put the figure showing the presence or absence of flagellate stages in the life cycle of green algae into supplemental information. In addition, please note that the ancestral state estimation analyses have been repeated based on newly inferred phylogenies including additional outgroup taxa, providing some new perspectives on the nature of the ancestral Viridiplantae.

5. The authors suggested that ILS was the main cause of inconsistent phylogenetic results. Is there any possibility that the authors could discuss more about ILS related to possible ancient radiation or adaptative evolution to global climates?

Reply: We have added some contents about ILS and possible ancient radiation to the discussion: “The occurrence of ILS across the short ancestral branch of Prasinodermophyta and Chlorophyta suggests that ancestors of green plants underwent an ancient radiation event, leading to rapid divergence into the major clades.”

Minor comments:

1. Line 79: “Proterocladus antiquus” -> “P. antiquus”

Reply: Done.

2. Line 122: “with fast-evolving genes removed” -> “without fast-evolving genes” .

Reply: Done.

3. Line 121-123: Please use the same term “long-branch artefact” or “long-branch attraction artifact” .

Reply: Done.

4. Line 132: “split in the Viridiplantae between Chlorophyta and Streptophyta” -> “split between Chlorophyta and Streptophyta” .

Reply: Done.

5. Line 136: “Pseudoscourfieldiales (clade V)” -> “Pseudoscourfieldiales” .

Reply: Done.

6. Line 203: e.g. -> e.g.,

Reply: Done.

7. Line 335: ” For the concatenation approach, The 576 OG alignments...” -> “For the concatenation approach, the 576 OG alignments...” .

Reply: Done.

8. Line 478, Line 482: the reference should be checked carefully, especially the journal and author names.

Reply: Done.

Reviewer #3

This manuscript adds to the body of evidence supporting Prasinodermophyta as sister to all other Viridiplantae groups. Understanding relationships across the green plant tree of life (and specially towards the root) is much needed to clarify, not just their role in key planetary events (e.g., great oxidation events during the Paleo- and Neoproterozoic), but also their evolution through time and how these events may have been caused by them or, instead, may have shaped them.

Reply: We greatly appreciate your valuable feedback. Currently, limited geological and geochemical studies have been conducted on the ancient Geologic time scale encompassing the Paleoproterozoic and Mesoproterozoic eras, primarily centered around oxygen levels. In our study, we have endeavored to establish a connection between oxygen levels and the origin and diversification of Viridiplantae. However,

further geological and geochemical evidence is required to delve into causal relationships.

Additionally, this manuscript sets an example on how to carefully disentangle systemic bias from biological signal, by presenting an in-depth exploration of biological mechanisms and systematic errors (methodological artifacts) underlying phylogenetic conflict and incongruence/discordance.

This manuscript also explores alternative timelines of events in early green plant diversification. The authors' constructive exploration of the effect alternative fossil constraints can have on ancestral divergence time estimation also adds to the literature trying to reconcile molecular clocks with the fossil and geochemical records.

For the most part, the authors' conclusions and claims are supported. I am missing, however, further exploration on the effect alternative outgroups can have in the ingroup topology, which could be having a major say on the long-branch attraction (LBA) detected (please, see further comments in the file attached).

In my opinion, further analyses with an extended outgroup that comprises Glaucophyta, Picozoa, and Rhodelphidia are needed before proceeding to accepting this work for publication (data can be mined from OA repositories). A major revision would be required, mostly because of the time repeating all of this tremendous work would take (with the extended outgroup mentioned above), not because of the quality of the analyses presented (which is excellent).

Reply: Thank you very much for your suggestion about the extended outgroups. We included Glaucophyta and Rhodelphidophyta into the outgroups in the revised manuscript, and have re-run all downstream analyses. We did not include the Picozoa for reasons explained below.

I would like to point out description of phylogenies could be further improved. Two clades (be them species poor or species rich) sharing a common ancestor (i.e., a node) are sister to each other (period). None of them is basal to the other, one just happens to have a lot less tips than the other (which could, for instance, result from much higher extinction rates,

rather than lower speciation ones). Please, be mindful of terms such as “basal”, “early”, “primitive”, etc., which can bias your readers’ perceptions of the evolution of these organisms.

Reply: We have removed the terms "basic" and "early", and reduced the use of "early" in the description of phylogenies in the revised version.

The methodology the authors present is sound and up to date (albeit the rudimentary character state reconstruction approaches implemented are rather limited on their reach). It meets the standards of the field; however, further work is needed to address the effect the outgroup chosen could be having on the LBA detected. I am also missing an extra step to remove outliers from gene alignments/topologies (see attached file for further details) prior to inferring species trees in a multispecies coalescent framework (it is known the quality of the input gene trees can greatly determine the output species tree topology). In all, there is sufficient level of detail to reproduce the authors’ approach (for those familiar with the methods used and who have certain degree of prior bioinformatic knowledge, that is).

Reply: Thank you very much for helpful comments and suggestions. In response, we have made extensive revisions in the revised version of the manuscript. Specifically, we have addressed your suggestions regarding the inclusion of extended outgroups, outlier removal, the utilization of analytical tools such as TreeShrink, AstralPlane, RAxML-NG, as well as other writing-related issues. In addition, we have incorporated the use of TreeShrink to identify outliers following the trimming step, removed these outliers from single-copy orthologous genes, then re-aligned and trimmed these genes to infer species trees.

Our specific responses to your comments in the manuscript are below:

1. Line 175-176: “we measured the level of gene tree conflict and ILS for the ancestral node of Chlorophyta and Streptophyta.” What about Prasinodermophyta?

Reply: This internal branch (node) was related with the position of Prasinodermophyta. The high proportion of conflicting bipartitions for the ancestral branch (node) of the Chlorophyta and Streptophyta indicated strong gene tree conflict regarding the

relationships among Prasinodermophyta, Chlorophyta and Streptophyta. In the revised manuscript, the high proportion of conflicting bipartitions for the ancestral branch of Prasinodermophyta and Chlorophyta indicated strong gene tree conflict regarding the relationships among Prasinodermophyta, Chlorophyta and Streptophyta.

2. Line 180: “With collapsed gene trees, quartet supports were calculated for three topologies surrounding this focal internal branch (T1, T2, and T3) in the coalescent analysis.” “With collapsed gene trees” What threshold? Why?

Line 205-206: “we collapsed branches below 30% to reduce gene tree estimation error” Mirarab (2019, DOI: 10.48550/arXiv.1904.03826), the creator of ASTRAL himself, recommends 10% (please, see comment in methods).

Line 332-333: “In order to reduce the influence of gene tree estimation error, branches in gene trees with bootstrap support below 30% were collapsed...” Why did you choose 30% as your bootstrap support collapse threshold? Mirarab (2019, DOI: 10.48550/arXiv.1904.03826), the creator of ASTRAL himself, recommends 10% as the cutoff value and points out collapsing above 50% can reduce accuracy substantially. Please, also see Simmons & Gatesy (2021, DOI: 10.1016/j.ympev.2021.107092), who recommend at least collapsing bipartitions with 0% bootstrap support.

Reply: Following your suggestions, in the revised manuscript, we selected 10% as bootstrap support collapse threshold to reduce gene tree estimation error for all related analyses.

3. Line 180: “With collapsed gene trees, quartet supports were calculated for three topologies surrounding this focal internal branch (T1, T2, and T3) in the coalescent analysis.” “Three topologies” represent quartets?

Reply: Yes. In the revised manuscript, three quartets surrounding the focal branch were represented with q1, q2, and q3.

4. Line 191: “we calculated RF distances for empirical gene trees and simulated gene trees.” Weighted or unweighted?

Reply: Unweighted. We made modifications in the revised manuscript:

“we calculated unweighted RF distances for empirical gene trees and simulated gene trees with high ILS conditions.”

5. Line 191-193: “The results showed that the coalescent model could account for 58% of gene tree variation observed for 339 gene trees (Supplementary Fig. 7).” Under which ILS conditions?

Reply: This was clarified in the revised manuscript: “The results showed that the coalescent model could account for 66% of gene tree variation observed for 434 gene trees under high ILS conditions.”

6. Line 197-198: “We, therefore, conclude that ILS contributed to nuclear-nuclear discordance concerning the position of the Prasinodermophyta.” You should mention Prasinodermophyta a lot sooner than at the very end of this paragraph. I have been wondering all along when you were going to bring this group up.

Reply: Thank you for your suggestions. In the revised manuscript, we have mentioned phylogenetic discordance regarding the position of Prasinodermophyta at the beginning of this paragraph.

7. Line 199-200: “Besides the presence of ILS, ancestral hybridization can similarly result in gene tree discordance and lead to incorrect species tree estimation.” What about the effects your outgroup of choice could have in the phylogenetic inference? Please, see methods for more detailed comments on this matter.

Line 209-211: “These results demonstrate the importance of exploring multiple possible causes of phylogenetic discordance in phylogenomic analyses, especially at deep nodes with a high level of gene tree variation.” Agreed. However, I am missing mention to the very stark effect your outgroup choice can have on the ingroup topology. Beyond the ingroup topology, your outgroup choice has implications for the character state reconstruction discussed next.

Line 214-215: “Our inferred phylogenetic relationships provided a solid framework for tracing key evolutionary events in early green plant lineages,” Maybe within the

ingroup, and with caveats. Your outgroup is too scarce to confidently assess ancestry towards the root.

Line 288-289: “To reconstruct reliable phylogenetic relationships, we have made attempts to reduce important sources of systematic bias (LBA)” The effect of the chosen outgroup on the ingroup topology should be further explored.

Line 321-323: “Taxon sampling of the plastid dataset included 60 plastid genomes downloaded from NCBI-GenBank, covering all classes of Prasinodermophyta, Chlorophyta, Charophyta, Bryophyta and two red algae as outgroups.” Provided you yourselves note LBA is at play, especially with regards to the outgroup, I find it very troubling only two red algae were chosen as said outgroup. What is the reasoning behind this choice? Why didn't you explore alternative outgroups that, beyond red algae, included glaucophytes (sister to Viridiplantae), Rhodelphidia (most probably sister to red algae), and Picozoa (most likely sister to the red algae plus Rhodelphidia clade)? The literature with regards to the effect the choice of outgroup can have on the ingroup is rather vast. For instance, see Rothfels et al. (2012, DOI: 10.1093/sysbio/sys001), Borowiec et al. (2019, DOI: 10.1016/j.ympcv.2019.01.024), or Aygoren Uluer et al. (2020, DOI: 10.1139/cjb-2019-0109), to mention just a few.

Line 395-396: “we reduced our taxon sampling to one outgroup and six ingroup taxa (two species of the Prasinodermophyta, two species of the Chlorophyta, two species of the Streptophyta, one species of the Rhodophyta).” Although you have immensely shrunk your ingroup, I am still concerned by your outgroup choice. Since Glaucophyta, and not Rhodophyta, have been inferred as sister to Viridiplantae time and again (e.g., 1KP).

Reply: We greatly appreciate your valuable suggestions regarding the outgroup selection.

Following your comments, we have made the following modifications:

For the nuclear dataset, we have added two species of Glaucophyta, one species of Rhodophyta, and one species of Rhodelphidophyta to the outgroup, re-conducted all the analyses, and updated the new results for the nuclear dataset. The quality of single-cell Picozoa genomes was found to be unsatisfactory. In most of gene alignments (>90%), Picozoa sequences had to be pruned and removed due to short sequence lengths. Therefore, we did not include Picozoa in the outgroup.

Given to the lack of chloroplast genomes for Rhodelphidophyta and Picozoa, we have added two species of Glaucophyta, and one species of Rhodophyta to the outgroup for the plastid dataset.

8. Line 349: “Detection of long-branch attraction” More importantly, are you familiar with TreeShrink? It allows the automated identification and removal of outlier long branches from phylogenies (greatly reducing the effect the presence of rogue taxa can have on phylogenetic inference; LBA inclusive). – Mai & Mirarab. 2018. TreeShrink: fast and accurate detection of outlier long branches in collections of phylogenetic trees. BMC Genomics 19 (Suppl 5), 272; DOI: <https://doi.org/10.1186/s12864-018-4620-2>.

Line 355-356: “ Suppression of putative long-branch artefacts were addressed in the following ways” I would suggest: (I) adding more taxa to the outgroup, (II) using TreeShrink on the gene data matrices required for the two-step coalescent approach (here RAxML + ASTRAL) to remove any putative outliers, and (III) concatenate those "shrunk" data matrices, with a larger outgroup and without outliers, for further analyses (here IQ-TREE). Also, word of caution, I would only trim alignments (here understood as excess gaps removal, i.e., trimAl) after (and not before) removing outliers (otherwise, you're dampening the noise said outliers make on the data matrices and their detection is stunted)

Line 397-398: “After aligning and trimming, 339 genes with a minimum of 1,00 aligned amino acid pairs and 100% taxa occupancy...” I would add an extra step between aligning and trimming for automated outlier removal with TreeShrink. Of course, you wouldn't necessarily end up with 100% taxa occupancy for all gene trees...

Reply: Thank you very much for your valuable suggestions. We have taken into account your comments (I)-(III) and re-analyzed our data accordingly. In our revised manuscript, the outliers were removed after the step of trimming. The reasons are as follows: First, sites with a higher number of gaps may indicate poor alignment quality and potentially contain erroneous phylogenetic signal. If gene trees are reconstructed before trimming, some erroneous phylogenetic signal may affect the accurate recognition and deletion of sequences with long branch lengths in each gene by Treeshrink. Conversely, by pruning

and removing these sites before tree reconstruction, we can reduce incorrect phylogenetic information and potentially mitigate the risk of long branch attraction. Secondly, we would like to highlight the work of Ma et al. (2021) (<https://doi.org/10.1038/s41467-021-26931-3>), who also used TreeShrink to detect and remove rogue taxa after the trimming step. Their approach supports our decision to identify outliers based on trimmed sequences as the ones that truly require removal.

9. Line 352-353 “The higher the long-branch score, the more susceptible the taxon is to long branch attraction.” Is there a threshold? How do you decide which is it?

Reply: In the previous version, the LB_score.py script was used to calculate long-branch score, but has no parameters to set a threshold. In the revised version, we used Treeshrink to detect long-branch attraction following your comments.

Line 360-362 “the average_branch_length.pl script (available at https://github.com/chrishah/phylog/blob/master/scripts/calculate_average_branch_length.pl) was used to calculate the average branch length of phylogenetic trees under different models.” Why not use the Median?

Reply: In the revised version, we used Treeshrink to detect long-branch attraction following your comments.

10. Line 365: “Dissection of phylogenetic signal” Are you familiar with R package treespace? Jombart et al. 2017. treespace: Statistical exploration of landscapes of phylogenetic trees. Mol Ecol Resour. 17: 1385–1392; DOI: <https://doi.org/10.1111/1755-0998.12676> As the title says, you can statistically (and visually) explore (plot) landscapes of phylogenetic trees (as long as said trees share its tips).

Reply: We have implemented TREESPACE to explore topological variability of individual gene trees based on the 7-taxon dataset. This additional analysis effectively visualizes three alternative hypotheses regarding the position of Prasinodermophyta.

11. Line 385-386: “the quartet frequency around each internal branch of the coalescent tree reconstructed...” Which can be further explored and visualized with DiscoVista or AstralPlane, in case these were not on your radar.

Reply: Thank you for your helpful recommendation, and in response, we have used AstralPlane to visualize the quartet frequency around each internal branch of the coalescent tree in the revised supplementary figures.

12. Line 390-392: “An ultrametric tree with branch lengths in mutational units (μT) was inferred with PAUP v 4.0a101 by constraining a ML tree search to the ASTRAL tree and using the concatenated alignment, LG+F+G substitution model, and enforcing a strict molecular clock.” This can also be done in RAxML-NG using the following parameters: `--evaluate --msa concat_alignment_here --model partition_scheme_here --tree ASTRAL_topology_here --brlen scaled` (see <https://github.com/amkozlov/raxml-ng>) The concatenation and partitioning can easily be implemented with AMAS (<https://github.com/marekborowiec/AMAS>).

Reply: Follow your advices, we have re-calculated the theta value using RAxML-NG with corresponding parameters: “The ML tree with branch lengths in mutational units (μT) was inferred with RAxML-NG by constraining a tree search to the ASTRAL tree topology and using a partition-by-gene scheme with best fit model for each partition (with main parameters -evaluate and -brlen scaled).” The methodology section in the revised manuscript is described as above.

13. Line 424: “Ancestral state reconstruction” As far as character reconstruction approaches go, the chosen ones are quite limited but not without value, although only for exploratory purposes. I would not put too much weight into inferences derived from the implementation of the Mk1 or the ER models. Much more nuanced character state reconstruction approaches exist (starting with the SSE family of methods, among many others).

Reply: Due to technical limitations, the approaches of reconstructing ancestral states cannot select the optimal model among all applicable models that fits the evolution of

traits, thus complex models may not be more suitable for traits such as cell morphology, habitat, and flagella stage than simple models. In addition, the complex model like BiSSE considers birth rate, death rate, and sample fraction. However, the sample fraction of our dataset is rather small, which possibly affect the accuracy of ancestral state estimation. Alternatively, we chose the ARD (all-rates-different) model which have more parameters than the standard Mk and ER model. In ARD model, every possible type of transition can have a different rate. The posterior probabilities of each character for each ancestral node were calculated from summaries of 1,000 simulations of stochastic character mapping under ARD model (the function make.simmap in Phytools). However, we approached the interpretation of the results obtained from the ancestral state estimation with caution, as explicitly stated in the manuscript.

14. Line 434: “The most clock-like 100 genes were identified from the dataset of” What were the chosen thresholds that led to this oddly even number of genes?

Reply: In our parallel investigation titled “Multiple independent evolution and Triassic-Jurassic origin of multicellular Volvocine algae” (under review), we explored the effect of different numbers of clock-like genes (100, 400, 700, 1000) on time estimations. The results revealed minimal disparities between estimates derived from the complete dataset and gene subsamples. Moreover, considering computational burden, we contend that selecting the 100 most clock-like genes for dating analyses is a viable approach.

15. Line 452: “Prasinodermophyta, Chlorophyta, Charophyta and Bryophyta” What about Tracheophyta?

Reply: Our analyses included the sampling of bryophytes; however, vascular plants were not included in our study. Our study really focused on the deep divergences of the Viridiplantae, therefore the inclusion of vascular plants will have no (or minor) effects on the general outcomes and conclusions.

16. Line 456: “Eight fossil calibration points were used for divergence time estimation.” I would appreciate an indication here (and not just in the main text) of why so few fossils made the cut.

Reply: We have added more explanation as follows: “Not only are there few records of green algal fossils, but some Precambrian green algal fossils with simple morphology have low taxonomic reliability. Thus, eight fossil calibration points were carefully selected for divergence time estimation.”

Reviewers' Comments:

Reviewer #1:

Remarks to the Author:

I appreciate the modifications included in this revised version and I am happy to recommend the publication of this manuscript.

Reviewer #2:

Remarks to the Author:

The manuscript (titled "Phylotranscriptomics unveil a Paleoproterozoic-Mesoproterozoic origin and deep relationships of the Viridiplantae")I reviewed has been thoroughly improved and is now in an excellent state.

After carefully reviewing the valuable comments and suggestions you provided on the previous version, I have examined each modification and made comprehensive revisions to the content, structure of the paper.

Through this process,

I am highly satisfied with the outcome of this revision and believe the manuscript is ready for further review and publication.

I kindly recommend editor to reassess the manuscript and consider accepting it for publication.

I believe the paper makes a significant contribution to the academic community and holds sufficient scholarly value.

Reviewer #3:

Remarks to the Author:

Dear authors, excellent job with the reviews. I have just found a typo in line 144: please, write LPP in capitals (rather than Lpp).

Also, thank you for bringing Ma J. et al. (2021; DOI: 10.1038/s41467-021-26931-3) to my attention. It was in my radar, but I had not checked their phylogenomic pipeline with enough care. I have explored shrinking long branches pre and post trimming with better results pre-trimming. I see this may be data matrix dependent and I hope, in the future, alternative shrinking strategies are further explored with simulated and real data (outside of the scope of your manuscript; just an observation for the record).

In my opinion, the current version of the manuscript is good to go. Looking forward to the published version.

REVIEWERS' COMMENTS

Reviewer #1 (Remarks to the Author):

I appreciate the modifications included in this revised version and I am happy to recommend the publication of this manuscript.

Reply: Thank you very much for your positive comments.

Reviewer #2 (Remarks to the Author):

The manuscript (titled "Phylotranscriptomics unveil a Paleoproterozoic-Mesoproterozoic origin and deep relationships of the Viridiplantae")I reviewed has been thoroughly improved and is now in an excellent state.

After carefully reviewing the valuable comments and suggestions you provided on the previous version, I have examined each modification and made comprehensive revisions to the content, structure of the paper.

Through this process,

I am highly satisfied with the outcome of this revision and believe the manuscript is ready for further review and publication.

I kindly recommend editor to reassess the manuscript and consider accepting it for publication.

I believe the paper makes a significant contribution to the academic community and holds sufficient scholarly value.

Reply: Thank you very much for your positive comments, and appreciation of our work.

Reviewer #3 (Remarks to the Author):

Dear authors, excellent job with the reviews. I have just found a typo in line 144: please, write LPP in capitals (rather than Lpp).

Also, thank you for bringing Ma J. et al. (2021; DOI: 10.1038/s41467-021-26931-3) to my attention. It was in my radar, but I had not checked their phylogenomic pipeline with enough care. I have explored shrinking long branches pre and post trimming with better results pre-trimming. I see this may be data matrix dependent and I hope, in the future, alternative shrinking strategies are further explored with simulated and real data (outside of the scope of your manuscript; just an observation for the record).

In my opinion, the current version of the manuscript is good to go. Looking forward to the published version

Reply: Thank you very much for your positive comments, and we have replaced the "Lpp" with "LPP".